# Speciation and Antibiotic Susceptibilities of Coagulase Negative Staphylococci Isolated from Ocular Infections

**DOI:** 10.3390/antibiotics10060721

**Published:** 2021-06-16

**Authors:** John E. Romanowski, Shannon V. Nayyar, Eric G. Romanowski, Vishal Jhanji, Robert M. Q. Shanks, Regis P. Kowalski

**Affiliations:** The Charles T. Campbell Ophthalmic Microbiology Laboratory, Department of Ophthalmology, University of Pittsburgh School of Medicine, Pittsburgh, PA 15213, USA; JER157@pitt.edu (J.E.R.); nayyars@upmc.edu (S.V.N.); romanowskieg@upmc.edu (E.G.R.); jhanjiv@upmc.edu (V.J.); shanksrm@upmc.edu (R.M.Q.S.)

**Keywords:** coagulase-negative staphylococci, eye infections, endophthalmitis, keratitis, conjunctivitis, blepharitis, API Staph, Biolog, DNA sequencing, *sodA* gene, antibiotic susceptibility

## Abstract

Coagulase-negative staphylococci (CoNS) are frequently occurring ocular opportunistic pathogens that are not easily identifiable to the species level. The goal of this study was to speciate CoNS and document antibiotic susceptibilities from cases of endophthalmitis (*n* = 50), keratitis (*n* = 50), and conjunctivitis/blepharitis (*n* = 50) for empiric therapy. All 150 isolates of CoNS were speciated using (1) API Staph (biochemical system), (2) Biolog GEN III Microplates (phenotypic substrate system), and (3) DNA sequencing of the *sodA* gene. Disk diffusion antibiotic susceptibilities for topical and intravitreal treatment were determined based on serum standards. CoNS identification to the species level by all three methods indicated that *S. epidermidis* was the predominant species of CoNS isolated from cases of endophthalmitis (84–90%), keratitis (80–86%), and conjunctivitis/blepharitis (62–68%). Identifications indicated different distributions of CoNS species among endophthalmitis (6), keratitis (10), and conjunctivitis/blepharitis (13). Antibiotic susceptibility profiles support empiric treatment of endophthalmitis with vancomycin, and keratitis treatment with cefazolin or vancomycin. There was no clear antibiotic choice for conjunctivitis/blepharitis. *S. epidermidis* was the most frequently found CoNS ocular pathogen, and infection by other CoNS appears to be less specific and random. Antibiotic resistance does not appear to be a serious problem associated with CoNS.

## 1. Introduction

Coagulase-negative staphylococci (CoNS) are normal inhabitants of the skin and mucous membranes [1]. Coagulase is a protein enzyme that, along with protein A, is bound to and associated with the *Staphylococcus aureus* cell wall. *S. aureus*, by itself, is a serious systemic pathogen of the skin, although there are many species of *Staphylococcus* that do not possess coagulase and are less pathogenic. There are over 45 species of CoNS [2].

Although part of the normal periocular flora [3], CoNS are considered opportunistic pathogens that cause endophthalmitis, keratitis, and conjunctivitis/blepharitis [4]. CoNS endophthalmitis is more common after cataract surgery because of the large load of bacteria inhabiting the eyelid margin [3,5]. CoNS keratitis may be less distinctive because of its association with normal flora, but an abundant number of colonies from corneal specimens obtained for laboratory studies indicate a possible pathogenic etiology [6,7,8,9]. CoNS as pathogens of conjunctivitis and blepharitis are not definitively diagnosed, due to a large presence on the eyelids [3], but cases have been described [9,10,11].

CoNS is generally not identified to the species level from eye cultures, mainly due to expediency. After the identification of *S. aureus* by the presence of coagulase and catalase, there are no practical tests to definitively determine the other staphylococcal species. Biochemicals have been utilized to speciate CoNS without much consistency in identification [6,7,8]. Pinna et al. speciated 55 CoNS (31 blepharitis, 12 conjunctivitis, 12 keratitis, and no endophthalmitis) into eight species using the commercial kit, API ID 32 Staph (bioMérieux, Paris, France), with more consistency, but there was no comparison with other identification methods [7]. Likewise, without comparison to other methods, Leitch et al. speciated *Staphylococci* from contact lenses using an identification system involving six biochemicals [12]. Their system was able to differentiate *Staphylococcus* into eight species, with a predominance of *S. epidermidis* and *S. capitis/warneri*. Monteiro et al. compared automatic identification (VITEK^@^ 2 system) with conventional methods (biochemicals) and genotypic identification (molecular analysis) of CoNS from blood samples. They found discrepancies within the three methods, but found a better correlation with the conventional methods and genotypic identification. They concluded that the more expensive automated system was more reliable in comparison to phenotypic identification for all bacterial isolates [13].

The first goal of the current study was to speciate CoNS using three methods: (1) API Staph (biochemical system), (2) Biolog GEN III Microplates (phenotypic substrate system), and (3) DNA sequencing of the *sodA* gene, from cases of endophthalmitis, keratitis, and conjunctivitis/blepharitis. The objective was to determine the correlation of *Staphylococcus* species with specific ocular infections. The second goal was to determine the susceptibility patterns of the different species of CoNS, to assure the efficacy of empiric treatment.

## 2. Results

Table 1 provides the identification of CoNS from endophthalmitis, keratitis, and conjunctivitis/blepharitis using API Staph, Biolog, and DNA sequencing. *S. epidermidis* at 80% (119 of 150) was the most prevalent CoNS species from ocular infections, as determined by the three identification systems. More species of CoNS were noted for conjunctivitis/blepharitis (13) and keratitis (10) than for endophthalmitis (6). Only 16% (24 of 150) of other CoNS isolates were identified with agreement among two or three methods. The Supplementary Information (Appendix A) contains the entire data set for the 150 isolates and the results of the three CoNS identification methods.

Table 2 shows the distribution of antibiotics used for the treatment of CoNS from endophthalmitis, keratitis, and conjunctivitis/blepharitis. For endophthalmitis, 100% of CoNS were susceptible to vancomycin and cefazolin. For keratitis, 100% of CoNS were susceptible to vancomycin and 98% were susceptible to cefazolin. For conjunctivitis and blepharitis, CoNS was not highly susceptible (30 to 82%) to any single antibiotic. Cefoxitin was not tested for CoNS conjunctivitis isolates.

Appendix A (Kowalski) is a supplementary table that contains the sequencing data for CoNS identification of ocular isolates.

## 3. Discussion

The virulence of CoNS as an opportunistic pathogen for ocular infections varies by the diagnosis. There is little doubt that CoNS, at 54% (372 of 684) (Campbell Laboratory data), is the most frequent cause of bacterial endophthalmitis, because the aqueous and vitreous contain no colonizing bacteria [4,14]. The implications of CoNS keratitis and conjunctivitis are supported clinically by the presentation of a large load of CoNS in corneal and conjunctival cultures. There are no distinct classical presentations of CoNS keratitis and conjunctivitis; both an inflamed eyelid margin from a blepharitis patient and a normal eyelid margin wsill present a positive culture for CoNS. Blepharitis is not generally infectious. CoNS is part of the normal flora for the eyelid margin; thus, it is difficult to implicate CoNS as the cause of inflammation. The role of CoNS in clinical blepharitis is based on the ophthalmologist’s impression and experience.

Treatment of CoNS ocular infections does not appear to be a therapeutic challenge. Methicillin resistance is not a problem for the treatment of ocular infections because there are effective alternatives for treatment. For endophthalmitis, prevention of CoNS infection is the real dilemma. A battery of topical povidone-iodine, topical antibiotics, and possibly an intracameral injection of antibiotics appears to be effective prophylaxis for most surgical cases [15,16]. Standard treatment of CoNS endophthalmitis is an intravitreal injection of vancomycin (1 mg) (200 µg/mL for a 5 mL vitreous volume). The half-life of vancomycin is 48 h in the inflamed human eye [17]. The present study indicates CoNS to be 100% susceptible to vancomycin.

In general, empiric infectious keratitis, which includes CoNS, is treated topically with fortified cefazolin (50 mg/mL) or vancomycin (20–50 mg/mL), and tobramycin (14 mg/mL) [18]. Fortified vancomycin (100%) and cefazolin (98%) both appear to be effective against CoNS, but both need to be formulated at a pharmacy. Commercially available 0.5% moxifloxacin is also used empirically to treat keratitis [19]. Our in vitro study indicates that moxifloxacin is less effective than vancomycin and cefazolin. The serum standard interpretation of CoNS susceptibility to moxifloxacin was 70% (35 of 50). The 30% resistance may be overreported due to high levels of moxifloxacin in the ocular tissue, which may be effective for treatment [20].

CoNS conjunctivitis is probably, but not definitely, self-limiting. Chronic conjunctival infections have been described with CoNS [10]. Generic antibiotics are generally used for the treatment of conjunctivitis/blepharitis because they are less expensive. Gram-positive topical antibiotics, with a conjunctivitis indication, such as polymyxin B/trimethoprim (82%), sulfacetamide (80%), and gentamicin (74%), may provide better coverage for acute infection. Cefoxitin has not been tested for CoNS conjunctivitis isolates. Beta-lactams are not used for conjunctivitis/blepharitis treatment. Blanco and Núñez indicated that moxifloxacin would provide coverage for both methicillin-susceptible and methicillin-resistant CoNS [21]. In contrast, Thomas et al. reported that the fluoroquinolone anti-infectives demonstrated decreased susceptibility for CoNS, but chloramphenicol (98.4% of 641 isolates) and tetracycline (82.4% of 176 isolates) provided better coverage [22]. It must be noted that if *S. haemolyticus* had not responded to polytrim (polymyxin B and trimethoprim), and was still believed to be a pathogen, the patient may have been placed on vancomycin.

Fortified vancomycin and cefazolin are excessive for CoNS conjunctivitis treatment and are not routinely tested. Blepharitis is generally treated topically with ointments that penetrate and remain longer on the eyelid margins. Bacitracin (78%), erythromycin (30%), and bacitracin/polymyxin B (82%) are sometimes cycled for blepharitis, which is often a chronic condition. The low susceptibility of CoNS to erythromycin (a bacteriostatic antibiotic) may be misleading because erythromycin is a cell-associated antibiotic [23,24,25]. It is more effective when attached to a cell wall than suspended in a broth. Macrolides can inhibit CoNS biofilm formation [26] and can act as anti-inflammatory agents against the chemotactic factors produced by neutrophils, which lead to eyelid inflammation [26,27,28].

The original goal of this study was to speciate CoNS and determine species correlations with ocular infections and in vitro susceptibility testing. There does not appear to be a practical and consistent method to definitively speciate CoNS in a timely manner for everyday identification. In contrast to the other two methods, API Staph identified eight CoNS isolates as *Staphylococcus aureus*; only one was identified by DNA sequencing and none by Biolog. All three methods were able to consistently speciate CoNS (80%) to *S. epidermidis*, but only 16% of CoNS were identified as other species. It must be noted that the manual system of Biolog was used instead of the more costly automated system. The manual system was used previously to speciate isolates of Moraxella [29]. Our study did not use MALDI-TOF-MS technology (matrix-assisted laser desorption/ionization-time of flight mass spectrometry), but, in a large-volume microbiology laboratory, CoNS identification to species may be improved using mass spectroscopy [30]. Unfortunately, as a small-volume laboratory, we did not have access to MALDI-TOF-MS, to identify CoNS as an additional comparison. Given the predominance of *S. epidermidis* among isolates and the high levels of susceptibility of CoNS to current antibiotics, a simple coagulase test still appears to be cost-effective and expedient, to distinguish *Staphylococcus aureus* from CoNS. Our study indicates that we need to find consistent methods to identify CoNS species in order to identify correlations with distinct clinical features of ocular disease.

The high concentrations of antibiotics delivered and directed toward ocular tissue are an advantage in the effective treatment of CoNS ocular infections. Antibiotics do not need to travel through the blood system to reach the target tissue. It is a common assumption in ophthalmology that adding an antibiotic directly to the infected site or injecting it into the vitreous provides optimal anti-infective therapy. The need to culture ocular infections and monitor the susceptibility of empiric antibiotics (e.g., vancomycin, cefazolin, moxifloxacin) will ensure future therapeutic success.

## 4. Materials and Methods

### 4.1. Coagulase-Negative Staphylococci

CoNS were cultured from patients presenting with endophthalmitis (*n* = 50), keratitis (*n* = 50), and conjunctivitis/blepharitis (*n* = 50) from a single tertiary medical center (University of Pittsburgh Medical Center, Pittsburgh, PA, USA). These cases were submitted for laboratory studies (The Charles T. Campbell Eye Microbiology Laboratory) with specific diagnoses designated on the patient requisition. The isolates were consecutively collected: endophthalmitis (August 2014 to July 2018), keratitis (May 2013 to November 2018), conjunctivitis/blepharitis (May 1998 to September 2018). The location of the culture (e.g., aqueous, vitreous, cornea, conjunctiva, eyelid) supported the diagnosis. Any CoNS growth from an endophthalmitis culture was considered significant as a pathogen, whereas 10 or more colonies on culture from the cornea or conjunctiva were necessary to suspect CoNS keratitis or conjunctivitis. The cut-off of 10 colonies was arbitrary and based on the senior author’s experience spanning over 40 years. (RPK). Normal conjunctiva and cornea flora, which includes the ocular surface, has no colonizing bacteria. Any collection of bacteria is generally around 1–4 colonies and probably comes from the eyelid margin. Manipulation by contact lens and administering topical drops could temporarily increase the contamination from the eyelid [12]. These areas are harsh environments for bacterial survival [3]. It must be noted that other reports indicated that 10 or more colonies on the conjunctiva and 100 or more colonies on the eyelid could be significant as pathogens [31,32]. The retrospective study did not require institutional review board/ethics committee approval because direct patient contact and personal information were not involved.

Endophthalmitis cultures were intraocular samples obtained from the aqueous and vitreous of the eye using a syringe and needle. The collected samples (a few drops) were routinely plated on trypticase soy agar supplemented with 5% sheep blood (SBA) (BBL™, Becton, Dickinson and Co., Sparks, MD, USA), aerobic chocolate agar (BBL™), anaerobic chocolate agar (BBL™), Sabouraud dextrose agar supplemented with gentamicin (BBL™), and an enriched thioglycolate broth (BBL™). A few drops of intraocular samples were placed on glass slides for direct examination by Gram and Giemsa stain to observe for microorganisms and cytology. For keratitis, the corneal scraping specimens were cultured directly, using spatulas or jeweler’s forceps to place the collected samples on SBA, aerobic chocolate agar, and Sabouraud dextrose agar supplemented with gentamicin. Collected samples were also placed on glass slides for direct examination by Gram and Giemsa stains to observe for microorganisms and cytology. Cultures of the conjunctiva and eyelid were collected with sterile soft-tipped applicators and placed on the same culture media as with keratitis (http://eyemicrobiology.upmc.com/PDFs/SpecimenCollection.pdf) (accessed on 26 February 2021).

As part of a clinical collection of bacteria for laboratory certification studies, bacterial growth on solid media was suspended in broth medium supplemented with 15% glycerol and stored at −80 °C. For this study, these isolates were retrieved by thawing and subculturing on SBA.

### 4.2. Antibiotic Susceptibility Testing of CoNS

Antibiotics are not only used to treat ocular infections, but also used prophylactically to prevent infections. Ophthalmologists use an array of fluoroquinolones, aminoglycosides, and other classes of antibiotics to treat bacterial infections. In this study, in vitro antibiotic susceptibilities of CoNS were determined using the disk diffusion method [33,34] on Mueller-Hinton II agar (BBL™). There are no susceptibility standards for the topical and intravitreal treatment of ocular infections. Susceptibility was interpreted using the CLSI (Clinical & Laboratory Standards Institute) serum standards; these are used to guide treatment without direct interpretation of susceptibility and resistance. It was assumed that the antibiotic concentrations in the ocular tissue are equal to or greater than the antibiotic concentrations attained in the blood serum.

In our clinical laboratory, routine antibiotic batteries are set up for both Gram-positive and Gram-negative bacteria. Cefoxitin is used to detect methicillin resistance in *Staphylococcus aureus* [33,34]. The antibiotic susceptibilities for CoNS were retrospectively determined from laboratory data used for laboratory certification. Antibiotics tested routinely for the treatment and prophylaxis of endophthalmitis were vancomycin, gentamicin, ciprofloxacin, ofloxacin, cefazolin, amikacin, ceftazidime, clindamycin, moxifloxacin, and cefoxitin. Antibiotics tested routinely for the treatment of keratitis were bacitracin, vancomycin, gentamicin, ciprofloxacin, ofloxacin, polymyxin B, cefazolin, tobramycin, sulfisoxazole, moxifloxacin, and cefoxitin. Antibiotics tested routinely for the treatment of conjunctivitis/blepharitis were bacitracin, erythromycin, gentamicin, ciprofloxacin, ofloxacin, trimethoprim, polymyxin B, tobramycin, sulfisoxazole, and moxifloxacin. Cefoxitin was not tested for CoNS conjunctivitis isolates, since beta-lactam antibiotics are rarely used for treatment.

It was not the intention of this study to recommend treatment or prophylaxis of CoNS ocular infection, but to confirm empiric therapy. Vancomycin is the standard empiric therapy for CoNS endophthalmitis; vancomycin or cefazolin is the standard empiric therapy for CoNS keratitis; conjunctivitis and blepharitis are treated with an array of different therapies based on the ophthalmologist’s preference.

### 4.3. API Staph

The CoNS were retrieved from frozen stocks by sub-culturing on SBA. The CoNS isolates were speciated by API Staph as directed by the package insert (https://www.mediray.co.nz/media/15784/om_biomerieux_test-kits_ot-20500_package_insert-20500.pdf) (accessed on 14 June 2021) (bioMérieux, Chemin de L’Orme, Marcy-L’Etoile, France).

### 4.4. Biolog

The CoNS were retrieved from frozen stocks by sub-culturing on SBA. Biolog GEN III Microplates (Biolog, Hayward, CA, USA) were used to identify CoNS according to the Biolog methodology (www.biolog.com) (accessed on 14 June 2021). In brief, the medium was inoculated with a CoNS isolate to a turbidity of 90% transmittance and aliquoted to a 96-well microplate at a volume of 0.1 mL per well. The plate was incubated at 34 °C and read manually for color changes at 6 h, 8 h, and 24 h. The tabulated data at each time point were entered into the Biolog Identification Systems Software (OOP 188rG Gen III Database v2.8). Species identification was determined as the most probable as indicated by the software.

### 4.5. DNA Sequencing

The CoNS were retrieved from frozen stocks by sub-culturing on SBA. The superoxide dismutase gene A (*sod*A) was the target gene for identifying CoNS [35]. This 429-bp-long DNA fragment encodes the manganese-dependent superoxide dismutase in 42 CoNS strains. Chromosomal DNA was obtained using QuickExtract™ DNA solution (Lucigen, Middleton, WI, USA), using the manufacturer’s protocol. Sequencing of the *sod*A gene was performed using degenerate primers following the protocol of Poyart et al. [35]. Primers were ordered from Integrated DNA Technologies (Coralville, IA, USA), and Taq DNA polymerase and reagents from New England Biolabs (Ipswich, MA, USA) were used. Sequencing was performed at the University of Pittsburgh Genomic Core facility and analyzed using NCBI BLASTN software [36]. The Supplementary Information (File S1. Kowalski DNA sequence Identification of CoNS) expands the description of CoNS identification by DNA sequencing.

Species were titled if BLASTN results yielded a percent identity over 90% and a high maximum ID score of 240 or greater. The sequences were compared to the other two identification methods for a corresponding match. Samples with poor quality sequence results were re-sequenced. The sequences were either a shorter length than required (~480 bp) or did not match in the BLAST database. Sequences with low similarity scores were sequenced at least twice to confirm the species identification.

## Figures and Tables

**Table 1 antibiotics-10-00721-t001:** Identification of coagulase-negative staphylococci (S.) from endophthalmitis, keratitis, and conjunctivitis/blepharitis using API Staph, Biolog, and DNA sequencing.

Isolated from Endophthalmitis	API Staph	Biolog	Sequencing	Correlation of ID Tests	
	*n* (%)	*n* (%)	*n* (%)	3 of 3	2 of 3
*S. epidermidis*	42 (84)	44 (88)	45 (90)	41	3
*S. hominis*	3 (6)	1 (2)	1 (2)	0	0
*S. lugdunensis*	2 (4)	4 (8)	3 (6)	0	3
*S. haemolyticus*	1 (2)	1 (2)	1 (2)	1	0
*S. capitis*	1 (2)	0 (0)	0 (0)		
*S. aureus*	1 (2)	0 (0)	0 (0)		
**Isolated from Keratitis**					
*S. epidermidis*	40 (80)	40 (80)	43 (86)	37	4
*S. caprae*	3 (6)	0 (0)	1 (2)	0	1
*S. hominis*	2 (4)	3 (6)	0 (0)	0	1
*S. warneri*	2 (4)	0 (0)	1 (2)	0	1
*S. lugdunensis*	1 (2)	1 (2)	1 (2)	0	2
*S. aureus*	1 (2)	0 (0)	1 (2)	0	2
*S. capitis*	0 (0)	4 (8)	2 (4)		
*S. pasteuri*	0 (0)	2 (4)	0 (0)		
*S. pettenkoferi*	0 (0)	0 (0)	1 (2)		
*Micrococcus* species	1 (2)	0 (0)	0 (0)		
**Isolated from** **Conjunctivitis/Blepharitis**					
*S. epidermidis*	31 (62)	33 (66)	34 (68)	28	6
*S. aureus*	6 (12)	0 (0)	0 (0)		
*S. haemolyticus*	2 (4)	3 (6)	5 (10)	2	0
*S. hominis*	2 (4)	2 (4)	2 (4)	0	2
*S. lugdunensis*	2(4)	4 (8)	2 (4)	1	2
*S. warneri*	2 (4)	1 (2)	2 (4)	0	1
*S. capitis*	1 (2)	3 (6)	1 (2)	1	0
*S. caprae*	1 (2)	1 (2)	2 (4)	0	2
*S. chromogenes*	1 (2)	0 (0)	0 (0)		
*S. cohnii*	1 (2)	1 (2)	1 (2)	1	0
*S. sciuri*	1 (2)	0 (0)	0 (0)		
*S. pasteuri*	0 (0)	1 (2)	1 (2)		
*S. saprophyticus*	0 (0)	1 (2)	0 (0)		

Correlation of ID tests is the number of identifications made by the 3 methods; 3 of 3 indicates that all methods had identical species IDs; 2 of 3 indicates that two methods had identical species IDs.

**Table 2 antibiotics-10-00721-t002:** Distribution of antibiotic susceptibilities (percent susceptible) for coagulase-negative staphylococci (CoNS). Identification determined by Biolog.

CoNS (Number Identified)											
**Endophthalmitis**	**VA**	**GM**	**CIP**	**OFX**	**CZ**	**AMK**	**CAZ**	**CC**	**MXF**	**FOX**	
*S. epidermidis* (44)	100	93.2	47.7	45.5	100	97.7	81.8	84.1	65.9	68.2	
*S. lugdunensis* (4)	100	100	100	100	100	100	100	100	100	100	
*S. hominis* (1)	100	100	100	100	100	100	100	100	100	100	
*S. haemolyticus* (1)	100	100	0	0	100	100	0	0	0	0	
Total (50)	**100**	94	52	50	100	**98**	**82**	84	68	70	
**Keratitis**	**BAC**	**VA**	**GM**	**CIP**	**OFX**	**PB**	**CZ**	**TOB**	**Sulfa**	**MXF**	**FOX**
*S. epidermidis* (40)	75	100	87.5	50	50	82.5	97.5	85	82.5	67.5	57.5
*S. capitis* (4)	100	100	75	75	75	100	100	100	100	100	100
*S. hominis* (3)	66.7	100	66.7	33.3	33.3	100	100	66.7	66.7	66.7	66.7
*S. pasteuri* (2)	100	100	100	100	100	100	100	100	100	100	100
*S. lugdunensis* (1)	0	100	100	0	0	100	100	100	100	0	0
Total (50)	76	**100**	86	52	52	86	**98**	**86**	86	70	64
**Conjunctivitis/** **Blepharitis**	**BAC**	**ERYT**	**GM**	**CIP**	**OFX**	**TMP**	**PB**	**TOB**	**Sulfa**	**MXF**	
*S. epidermidis* (34)	79.4	26.5	70.6	47.1	47.1	50	85.3	70.6	79.4	29.4	
*S. lugdunensis* (3)	66.7	0	100	66.7	66.7	33.3	100	100	66.7	66.7	
*S. hominis* (2)	100	0	100	50	50	0	50	100	100	50	
*S. haemolyticus* (3)	66.7	0	33.3	0	0	0	100	0	33.3	0	
*S. cohnii* (1)	0	0	100	100	100	100	100	100	100	100	
*S. saprophyticus* (1)	0	0	100	100	100	100	0	100	100	100	
*S. capitis* (3)	100	100	100	100	100	100	100	100	100	66.7	
*S. pasteuri* (1)	100	100	0	100	100	0	0	0	100	0	
*S. warneri* (1)	100	100	100	0	0	100	100	100	100	0	
*S. caprae* (1)	100	0	100	0	0	0	0	100	100	100	
Total (50)	**78**	**30**	74	50	50	**48**	**82**	72	**80**	36	

AMK: amikacin; BAC: bacitracin; CIP: ciprofloxacin; CZ: cefazolin; CAZ: ceftazidime; CC: clindamycin; ERYT: erythromycin; FOX: cefoxitin; GM: gentamicin; MXF: moxifloxacin; OFX: ofloxacin; PB: polymyxin B; Sulfa: sulfisoxazole; TMP: trimethoprim; TOB: tobramycin; VA: vancomycin. Susceptibility was interpreted using the CLSI (Clinical & Laboratory Standards Institute) serum standards. It is assumed that the antibiotic concentrations in the ocular tissue are greater than the concentrations in the blood serum. **BOLD** indicates empiric antibiotics.

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
