# Peer review of "Speciation and Antibiotic Susceptibilities of Coagulase Negative Staphylococci Isolated from Ocular Infections"

_antibiotics, 2021, doi:10.3390/antibiotics10060721_

Round 1

Reviewer 1 Report

This is a study with the objective of to identify CoNS species isolated from patients with ocular infections and the profile of resistance to antimicrobials. The article is interesting but needs major revision.

Abstract: line 24; This is not true; it may not be so evident among CoNS isolates from ocular infections, but  CoNS isolated from other clinical materials and other infections we have greater resistance to anticrobials than S. aureus.

And their results show a 30% resistance to beta-lactam antibiotics, in addition to being evident that some species like S. haemolyticus show a multi-resistance. This needs to be highlighted in its conclusion and abstract.

Introduction: lines 43-46; Explain better about the role of coagulase. There are two types of coagulase, the one bound to the cell wall, called the “Clumping Factor” and the free coagulase which is the other standard method for differentiating S. aureus from CoNS in infections in humans.

Describe more about the various species of CoNS that may be associated with ocular infections based on recent articles and also about the differences in relation to the antimicrobial resistance profile, which reinforces the importance of speciation of CoNS.

Lines 60-65; Change that paragraph. It gives the impression that API ID 32 Staph is better than traditional biochemical tests for identifying CoNS and this is not true (Cunha at al. 2004; https://doi.org/10.1590/S0074-02762004000800012).

The traditional biochemical method is still the other standard in the identification of CoNS species, although it is laborious and has been routinely replaced by faster and easier to perform methodologies. Some rapid methods show several problems in the identification of CoNS, with discrepant results and serious errors such as identifying S. aureusas CoNS or the other way around. I suggest write more about the limitations these methodologies.  (Cunha at al. 2004; https://doi.org/10.1590/S0074-02762004000800012, Monteiro et al. 2016. DOI 10.1186/s12941-016-0158-9).

Results and Methodology:  Confirm the identification of the Staphylococcus genus by the fermentation / oxidation method or by genotypic identification from the research of the RNAr 16S gene for the isolates that were sensitive to bacitracin. Staphylococcus are differentiated from Micrococcus species on the basis of the oxidation and fermentation of glucose, resistance to bacitracin (0.04 U) indicated by absence of an inhibition halo or presence of an inhibition halo measuring up to 9 mm in diameter, and susceptibility to furazolidone (100 µg) characterized by inhibition zones measuring 15 to 35 mm in diameter (Baker 1984, Comparison of various methods for differentiation of staphylococci and micrococci.  J Clin Microbiol 19: 875-879).  I suggest doing the test of the discs too. Koneman, E. W., Allen, S. D., Janda, W. M., and Schreckenberger, W. C. W. (1997). Color Atlas and Textbook of Diagnostic Microbiology, 5th Edn. Philadelphia, PA: Lippincott.

The authors need to explore further the result of cefoxitin (FOX), as this test detects CoNS resistant to methicillin. And as shown in Table 3, 30% of CoNS are resistant to methicillin, this data being very important, to differentiate the multiresistant isolates, which is the case of S. haemolyticus and some isolates of S. epidermidis.

The results of the cefoxitin test need to be extrapolated to the cefazoline result, as both are beta-lactams antimicrobials and although it is sensitive to cefazolin in vitro will not respond in vivo, as it probably carries the mecA gene that confers resistance to everyone beta-lactams, including cephalosporins and carbapenems, with the exception of only fifth generation cephaloporins (Ceftobripole and Ceftaroline) - CLSI, 2020. It would be very important for the quality of the work to include the search for mecA gene, for resistance to methicillin.

I suggest improving the tables and include a table with CoNs resistant to methicillin and those sensitive to methicillin, showing the number of resistances to other tested antimicrobials.

Discussion: lines 230-233; Not only due to the predominance of S. epidermidis, but the coagulase test is essential for the differentiation of S. aureus from CoNS and from there to do the test for the differentiation of CoNS species. It is wrong to stop the identification only in the coagulase test, because although S. epidermidis is the most frequent species, it is not the only one involved in cases of ocular infections, and the resistance to antimicrobials is very different, as can be verified for S. haemolyticus.

Minor Changes:

Change CNS for CoNS, this is the better abbreviation for coagulase-negative staphylococci.

S. lugdunensis and not S. lugdunesis.

Table 1. Improve the table and the layout. Exclude % of all rows and place them under the subtitle in the columns: API Staph

                                                                       N (%)

Author Response

please see attachment in the box

Reviewer 2 Report

This manuscript by J.E. Romanowski et al. describe very interesting data on the CNS isolated from ocular infections.

Even if this manuscript deserve publication, extensive revision is needed before acceptance.

Introduction : 

Why have the authors not considered MALDI-TOF identification for this study ? Is it not possible to retrospectively identify these strains using this method?

bioMérieux is not located in Montalieu Vercieu ... verify.

Results : 

More than table, Venn's Diagramm would be very useful.

Moreover, how the authors considered verification of discrepant results ?

Line 167-169 has to be included in the methods part of the manuscript.

Discussion : 

Have the authors considered to discuss the absence of objective criteria for the role of CNS in clinical blepharitis?

Methods :

Why have the authors considered a cut-off of 10 considered as a pathogen?

How was the endophthalmitis culture been sampled?

Commercial culture media have to be correctly referenced

Website could be reference as other reference.

Cut-off values for AST determination have to be referenced.

Have the sequence been deposited on a publicy-available database? If yes, please give accession number. If not, please perform.

Global :

"e.g." has to be italicized

numerous typos remain (bold for example).

Round 2

Reviewer 1 Report

My comments remains the same as in the first review. My suggestions were not accepted and not even justified for not being accepted.

Author Response

Please attached file

Reviewer 2 Report

Even if some limitations remains, these could not be addressed due to the design of the study.

We could regret the absence of MALDI-Tof verification, that could be considered in a multicenter study; and the threshold of 10 already described in my previous comment.

Except for this major limitation, the manuscript is suitable for publication and so I would let the editor decide the conclusion for this manuscript.
